# Response Evaluation after Neoadjuvant Chemotherapy for Resectable Gastric Cancer

**DOI:** 10.3390/cancers15082318

**Published:** 2023-04-15

**Authors:** Alina Desiree Sandø, Reidun Fougner, Elin Synnøve Røyset, Hong Yan Dai, Jon Erik Grønbech, Erling Audun Bringeland

**Affiliations:** 1Department of Gastrointestinal Surgery, St. Olavs Hospital, Trondheim University Hospital, 7030 Trondheim, Norwayerling.audun.bringeland@stolav.no (E.A.B.); 2Department of Clinical and Molecular Medicine, Faculty of Medicine and Health Sciences, NTNU—Norwegian University of Science and Technology, 7034 Trondheim, Norway; 3Department of Radiology, St. Olavs Hospital, Trondheim University Hospital, 7030 Trondheim, Norway; 4Department of Pathology, St. Olavs Hospital, Trondheim University Hospital, 7030 Trondheim, Norway; 5Clinic of Laboratory Medicine, St. Olavs Hospital, Trondheim University Hospital, 7030 Trondheim, Norway

**Keywords:** gastric cancer, neoadjuvant chemotherapy, response evaluation, RECIST, downsizing, TNM, downstaging

## Abstract

**Simple Summary:**

Response evaluation following neoadjuvant chemotherapy for gastric cancer has been widely debated. Following a full course of preoperative cycles, response evaluation can guide the vigor in delivering postoperative chemotherapy, indicate for whom a shift in regimen following a later relapse may be prudent, and be a tool for scientific use. In the everyday setting, the terms downsizing and downstaging are often used synonymously with response to treatment, but any abilities and limitations in this respect need to be proven. The aim of the present study was to investigate downsizing and downstaging as methods of response evaluation following NAC in gastric cancer. We evaluated whether the response mode translated into strata of long-term survival rates, a prerequisite for a useful method in a neoadjuvant setting.

**Abstract:**

Background: The method of response evaluation following neoadjuvant chemotherapy (NAC) in resectable gastric cancer has been widely debated. An essential prerequisite is the ability to stratify patients into subsets of different long-term survival rates based on the response mode. Histopathological measures of regression have their limitations, and interest resides in CT-based methods that can be used in everyday settings. Methods: We conducted a population-based study (2007–2016) on 171 consecutive patients with gastric adenocarcinoma who were receiving NAC. Two methods of response evaluation were investigated: a strict radiological procedure using RECIST (downsizing), and a composite radiological/pathological procedure comparing the initial radiological TNM stage to the pathological ypTNM stage (downstaging). Clinicopathological variables that could predict the response were searched for, and correlations between the response mode and long-term survival rates were assessed. Results: RECIST failed to identify half of the patients progressing to metastatic disease, and it was unable to assign patients to subsets with different long-term survival rates based on the response mode. However, the TNM stage response mode did achieve this objective. Following re-staging, 48% (78/164) were downstaged, 15% (25/164) had an unchanged stage, and 37% (61/164) were upstaged. A total of 9% (15/164) showed a histopathological complete response. The 5-year overall survival rate was 65.3% (95% CI 54.7–75.9%) for TNM downstaged cases, 40.0% (95% CI 20.8–59.2%) for stable disease, and 14.8% (95% CI 6.0–23.6%) for patients with TNM progression, *p* < 0.001. In a multivariable ordinal regression model, the Lauren classification and tumor site were the only significant determinants of the response mode. Conclusions: Downsizing, as a method for evaluating the response to NAC in gastric cancer, is discouraged. TNM re-staging by comparing the baseline radiological CT stage to the pathological stage following NAC is suggested as a useful method that may be used in everyday situations.

## 1. Introduction

Three European randomized controlled trials (RCTs) have claimed improved long-term survival rates for patients with resectable gastric cancer with the addition of perioperative chemotherapy to surgery [1,2,3]. A common denominator among them is their broad inclusion criteria, in terms of both demographic variables and tumor characteristics. Evidence suggests that there can be a differential response to the neoadjuvant part of chemotherapy (NAC) [4], although, as of yet, there are no clinicopathological variables established to serve as guidance for an individually tailored treatment [5,6,7,8].

The optimal method of response evaluation following NAC is still a matter of debate [9,10]. Two main approaches are available: a histopathological one, such as the Mandard or Becker scores [11,12], or a radiological one, based on computed tomography (CT) [13,14]. Metabolic markers for response evaluation, such as FDG PET-CT, are promising, but they are only at an exploratory stage at present [15,16,17]. Limitations of the histopathological methods include restriction to the primary tumor, discarding information from lymph nodes or metastatic deposits, and that patients rendered medically unfit for surgery following NAC are not part of the evaluation. Further, standardized interpretations of the histological findings are required [18]; so far, conflicting results have been reported on the correlation between the histopathological response and long-term survival rates [19,20,21,22,23,24].

For radiology, two main approaches exist. The first is quantifying the response by measuring any change in tumor size, with the response evaluation criteria in solid tumors (RECIST) being the most frequently used tool [13]. This method was designed for evaluating solid tumors in a non-curative setting, but it is frequently encountered in scientific reporting on the response to NAC in gastric cancer [9,25,26,27,28,29]. It is of great importance to disclose any limitations of downsizing as a concept of the response to NAC in a Western cohort of resectable gastric cancers. The second is the response evaluation in terms of downstaging, either as a strict radiological procedure or as a composite radiological/pathological procedure. The former, comparing the radiological stage before chemotherapy (rTNM) to the radiological stage after chemotherapy (yrTNM), has been proven to be unreliable due to the notoriously low accuracy of post-NAC CT staging, with oedema and fibrosis impeding the proper assessment of the T and N categories [14,30]. To circumvent this problem, a composite method of downstaging is suggested, where the radiological stage at diagnosis (rTNM) is compared to the pathological stage following chemotherapy (ypTNM). Published data on this approach to response evaluation in gastric cancer are scarce [31,32,33].

A prerequisite for a useful method of response evaluation after NAC in upfront resectable cancer is the ability to stratify patients into groups of different long-term survival rates based on the response mode. However, improved long-term survival rates associated with the local tumor response to NAC could merely be a matter of confounding, apprizing as responders those tumors with a favorable prognosis from the outset.

The aims of the present study were, first, to investigate the potentials and limitations of the radiological approaches in response evaluation to NAC in everyday situations, using already- available radiological and pathological information. The second aim was to address any confounding between the response mode and long-term survival rates by analyzing the association stratified on the basis of baseline rTNM stages.

## 2. Materials and Methods

### 2.1. Study Design

This was a population-based, retrospective study conducted in Central Norway, with some 700,000 inhabitants. Patients diagnosed with gastric cancer from 1 January 2007 to 31 December 2016 were identified through the Norwegian Cancer Registry (NCR). Evaluation and treatment were centralized to St. Olavs Hospital, the university hospital of Central Norway. By reviewing individual electronic patient journals, the registration of relevant clinical variables for 733 consecutive patients with gastric adenocarcinoma was obtained, including tumors of the gastric cardia Siewert types II/III. Of these, 171 patients met the criteria for perioperative chemotherapy as part of the national standard, which has been in place since January 2007, and were the objective of the present study.

### 2.2. Staging and Treatment

Initial staging included CT and gastroscopy. Endoscopic ultrasound, PET-CT, and diagnostic laparoscopy were not a part of the standard assessment, in accordance with the national guidelines at the time. Radiologic staging was performed using a multidetector CT (Siemens Somatotom Definition Flash or AS+, with detector 128 × 0.6) with an iterated scan following NAC. The procedure was standardized with a protocol offering optimal gastric distention [34]. Images were obtained after 45 and 70–75 s. The image volume was reconstructed to series of 1.5 and 3 mm thin-slice images in the axial, coronal, and sagittal planes. For study purposes, a senior gastro-radiologist blinded to the final ypTNM status revisited all the CT evaluations.

Following national guidelines, patients aged ≤75 years, with a WHO performance status of 0–1, and of disease stages Ib-III according to the UICC 7th edition were offered a MAGIC-style regimen of perioperative chemotherapy, consisting of i.v. epirubicin 50 mg/m^2^ on Day 1, i.v. oxaliplatin 130 mg/m^2^ or i.v. cisplatin 60 mg/m^2^ on Day 1, and oral capecitabin (Xeloda^®^) 1250 mg/m^2^ for 21 days (EOX/ECX), with three cycles prior to surgery and three to follow for the radically resected. Surgery was, by default, a modified D2 dissection, with intraoperative frozen sections routinely obtained. The censoring date was 31 December 2021, providing a minimum follow-up of 5 years. This study was approved by the Regional Ethics Committee, and the manuscript was prepared in accordance with the STROBE guidelines [35].

### 2.3. Downsizing

The RECIST 1.1 criteria were used to evaluate the response in terms of a change in metric size following NAC. According to RECIST, target lesions are divided into non-nodal (measurable tumors ≥10 mm) or nodal (lymph nodes with short axis ≥15 mm). The sum of the diameters, the longest for non-nodal lesions and the short axis for nodal lesions, was calculated at the baseline CT and at restaging CT following NAC. Analyses were conducted with three response categories according to RECIST. 1: Response, defined as at least a 30% decrease in the sum of the diameters of the target lesions. 2: Progression, defined as at least a 20% increase in the sum of the diameters of the target lesions, and an absolute increase of at least 5 mm. The appearance of any new lesions (M+ and/or pathological lymph nodes) was also considered progression. 3: Stable, otherwise.

### 2.4. Downstaging

The Union for International Cancer Control (UICC) TNM classification 7th edition was used for baseline CT staging (rTNM). At radiological staging, the T stage was set as follows: T1, invisible tumor or focal thickening of the mucosal layer; T2, focal thickening of the gastric wall with a smooth outer border; T3, diffuse or focal transmural thickening of the gastric wall with a blurry border to periventricular fat; T4, tumor infiltrating the serosal lining or adjacent organs. Lymph nodes were judged as positive when the short axis was ≥10 mm or they were exhibiting a pathological structure. For the pathological examination of the resected specimen, the stage was reported as ypTNM according to the UICC 7th edition. Response evaluation was performed by comparing the radiological stage at baseline (rTNM) to the pathological stage following NAC (ypTNM). Analyses were conducted with three response categories: response (downstaging, e.g., stage IIIc→IIIa), stable disease, and progression (upstaging, e.g., stage IIa→IIIb).

### 2.5. Statistics

Continuous variables are reported as the median and range. Categorical variables were cross-tabulated and analyzed using the χ^2^ statistic. For TNM downstaging, an ordinal multivariable regression was performed with the response category to NAC as the dependent variable with three categories, using a cumulative logit model. Gender, age (continuous), and variables with a *p*-value of ≤0.20 from the univariable χ^2^ analyses were taken as the explanatory variables. The proportional odds assumption was verified (test of parallel lines). Overall survival was counted from the time of diagnoses, computed by the Kaplan–Meier method, and compared using the log-rank test. The level of statistical significance was set at *p* = 0.05. Analyses were performed using IBM SPSS Statistics version 27.

## 3. Results

A total of 171 patients with a median age of 63 years (range 27–77) initiated perioperative chemotherapy, with 57 (33%) receiving ECX and 114 (67%) receiving EOX. Tolerability was comparable to that reported in other studies, with 145/171 (85%) able to complete all three preoperative cycles. Of these, 98/171 (57%) commenced postoperative chemotherapy, and 71/171 (42%) completed all six cycles [1,3]. Following NAC, 18/171 (11%) did not receive surgery (1 patient wished not to, 5 were rendered medically inoperable, and 12 had metastatic disease at CT restaging). Of the 153 patients receiving surgery, 140 were radically resected, 10 had a R2 resection, and 3 underwent merely an explorative laparotomy.

The 5-year overall survival for the entire cohort was 42.0% (95%CI 34.6–49.4%). In all, 2 patients did not have restaging CT, 1 died of unrelated causes during NAC, and 1 refused further imaging and treatment, leaving 169 patients accessible for response evaluation.

### 3.1. Response Evaluation by Downsizing

Of the 169 patients, 18 had a nodal target lesion to measure, and the remaining 151 underwent an assessment of response based on the longest diameter of the gastric primary, as suggested by the RECIST response group [36]. Overall, 60/169 (36%) patients were responders, 4 with a complete response; 92/169 (54%) had stable disease; and 17/169 (10%) had progressed. Of those progressing, 5 did so by a ≥20% increase in the sum of the diameters, whereas for 12, peritoneal carcinomatosis was detected in the restaging CT, classifying these patients as having progressive disease according to RECIST. No patient developed deep organ metastases. CT following NAC failed to identify a further 12 patients with peritoneal carcinomatosis which was only acknowledged at operation, leading RECIST to misclassify 4 of these as responders and 8 as being of the stable disease category. The response mode was not able to stratify patients into groups of different long-term survival rates (Figure 1). Unsurprisingly, patients classified with progression had an inferior survival rate since the majority (71%) were allocated to this subgroup due to metastatic disease. No difference in long-term survival rates was found when comparing the group with stable disease to those responding (log-rank *p* = 0.570).

### 3.2. Response Evaluation by Downstaging

Following NAC, 26 patients were found to have metastatic disease—either by restaging CT (*n* = 12), by laparotomy (*n* = 12), or by pathology (*n* = 2)—and these patients could have ypTNM stage IV disease assigned in spite of not all being resected. In total, 5 patients with M0 disease following NAC did not reach surgery and could not have a pathological stage assigned, meaning that there were 164 patients available for response evaluation by TNM restaging. Of these, 78/164 (48%) were downstaged, 25/164 (15%) were of stable disease, and 61/164 (37%) were upstaged (Figure 2). A histopathological complete response was found in 15/164 (9%) of the patients. Of the patients classified with disease progression, 26/61 (43%) progressed by developing M+ disease (Figure 2), all with advanced T3 or T4 tumors at the outset. No tumors staged ≤T2 progressed to metastatic disease following NAC.

The TNM response mode did achieve the objective of stratifying patients into groups of significantly different long-term survival rates (Figure 3A). The 5-year overall survival rates were 65.3% (95% CI 54.7–75.9%) for the responders, 40.0% (95% CI 20.8–59.2%) for stable disease, and 14.8% (95% CI 6.0–23.6%) for disease progression, with a global log-rank *p*-value of ˂0.001. Subdividing patients with disease progression into those maintaining the M0 status versus those progressing to M+ disease, survival in the M0 subset was similar to that of patients with stable disease (*p* = 0.512) (Figure 3B). Survival for the responders was further compared to that for the merged group (stable disease + progression maintaining M0), with an improved survival rate for the responders as a sustained finding across the individual baseline rTNM stages (Figure 4A,B) and graded according to the number of tiers downstaged (Figure 4C).

By univariable analysis, the Lauren category (*p* < 0.001) was the only significant determinant of the response mode, with tumor stage (*p* = 0.065) and tumor location (*p* = 0.097) having a non-significant association (Table 1). This was contingent upon a particularly high rate of progression for the Lauren diffuse tumors (56%) and tumors of an anatomic diffuse location (68%), predominantly to M+ disease. The advanced disease stages IIIB and IIIC showed particularly high rates of response of 63% and 65%, respectively (Table 1, Figure 2). In a multivariable ordinal regression model, the Lauren category and preoperative tumor location remained the only significant determinants of the response mode. The model fitted the data well, with Pearson’s overall goodness of fit *p* = 0.17 and pseudo R^2^ (Nagelkerke) = 0.28 (Table 2).

## 4. Discussion

When discussing response evaluation to chemotherapy, disease context is important. In a palliative situation, the primary aim is to relieve pain or hollow organ obstruction, and attending downsizing is reasonable. In a neoadjuvant setting with resectable tumors, improving long-term survival rates is the goal, and any useful method of response evaluation must be able to stratify patients into groups of different long-term survival rates. The present study is the first to detail how RECIST performs in a Western population with upfront resectable gastric cancers. Only 10% of the patients had a nodal target lesion to evaluate. The remaining 90% were confined to measuring the longest tumor diameter, as suggested by the RECIST working group [36], with the concern of this varying with the degree of gastric distention at CT and assessment compromised by tumors frequently stented prior to CT re-evaluation. The method was unable to assign patients to subsets with different long-term survival rates (Figure 1), corroborating findings from some Asian studies [9,21,37]. This correlates to the fact that the concordance between a change in longitudinal diameter and a change in perpendicular thickness as a surrogate marker of T stage is known to be weak in the present paper with a ƙ = 0.5 (95% CI 0.36–0.64), and no prognostic association is established between tumor diameter and long term survival rates [38,39]. A further shortcoming resides in the low sensitivity of CT to detect peritoneal carcinomatosis [40]. Of the 10,000 patients constituting the evidence base for RECIST, no trials on gastric cancer patients or NAC were included [41]; still, the method is persistently used for this purpose even today [27,28,42]. We argue that this inappropriate use should be abandoned.

Downstaging as a marker of improved long-term survival rates needs to be proven. Arguably, it could merely represent a cosmetic finding unable to account for chemoresistant micro-metastases. We recently demonstrated that evaluating strict radiological downstaging by comparing the CT stage before chemotherapy (rTNM) to that of restaging CT (yrTNM) is highly unreliable and should be discouraged [14]. In the present study, the aim was to compare the initial radiological stage (rTNM) to the pathological stage following NAC (ypTNM), circumventing the notoriously low accuracy of CT staging following chemotherapy [30,43]. This approach did meet the requirement for the response mode to translate into strata of significantly different long-term survival rates (Figure 3 and Figure 4), and it complies with the intuitive notion of improved survival following downstaging. These findings are novel, as published data on this approach to response evaluation are scarce [31,32,33,44]. In a multivariable ordinal regression model, the tumor location and Lauren histological type were significant determinants of the response mode, whereas gender, age, and rTNM stage were not. Although advanced disease stages did respond well to NAC, they also harbored the tumors that progressed to metastatic disease, balancing out any statistical significance. Some of these may even have had metastatic disease unrecognized at the initial CT staging, supporting the stance that diagnostic laparoscopy should be a part of the initial workup when facing advanced disease [45].

Regarding MSI status, a previous study found an inferior survival rate for the MSI-H group receiving chemotherapy compared to those receiving upfront surgery, indicating chemo-resistance for this subset [8]. In the present study, only 8/164 (5%) tumors were MSI-H, which is too few to draw any conclusions.

The overall 5-year survival rate for the responders was 65.3%. The group included a high proportion of patients with advanced disease (Table 1), indicating that chemotherapy did not merely act to select as responders those patients with a favorable prognosis from the outset, but exerted a true beneficial effect. This sentiment is further strengthened by examining the survival vs. response mode, stage by stage (Figure 4A–C). The group of patients with stable disease was small (15%), indicating that NAC preferentially induces a response or allows progression to take place. Notably, the 5-year overall survival for patients with stable disease was 40.0%, which is on par with that reported from historic cohorts operated prior to the introduction of NAC [5,39,46], and similar to that of patients with disease progression but maintaining their M0 status (Figure 3B). Although the MAGIC regimen was recently replaced by the FLOT4 regimen as the standard of care in the West [3], both are based on elements of traditional chemotherapy and delivered with a “one-size-fits-all” approach. This underscores the need to identify markers for subsets likely to enjoy a survival benefit from perioperative oncological treatment. The method of composite TNM downstaging may be a valuable tool in this process. Ideally, response evaluation could be attempted at an earlier stage to decide on whether to continue NAC or move on to early surgery. However, the fixed number of cycles delivered for both regimens is small, making an interval assessment difficult, as proven by a recent Dutch study [47]. An inadequate amount of time may have elapsed for substantial CT changes to manifest themselves. Adding to this, the accuracy of CT staging following chemotherapy is poor, as previously stated [14]. Rather, the methods of response evaluation following a full course of preoperative cycles must be valued as tools for scientific use, to guide the vigor in delivering postoperative chemotherapy, and to signal for whom a shift in regimen following a later relapse may be prudent.

A limitation of this study is the inherent challenge of accurate baseline CT staging, especially for the N category. For chemo-naïve patients, the concordance between CT staging and pathological staging is in the range of 69–88% for the T category and 51–71% for the N category; neither EUS, nor MRI, nor PET could establish the N category with any higher precision [48,49]. Lowering the node diameter threshold beyond 10 mm would only significantly reduce specificity. However, in the present study, CT stage as a proxy for the initial pathological stage, paired to the pathological stage following chemotherapy, indeed met the goal of stratifying patients into groups of different long-term survival rates based on the response mode. A strength of the present study is that it is population-based and spans a decade of inclusion. The preoperative workup, chemotherapy delivered, and surgery were all standardized. It is considered favorable that all CT and pathological examinations were performed at the same institution by a dedicated gastro-radiologist and gastro-pathologists, respectively.

## 5. Conclusions

*Downsizing*, as a method of response evaluation following NAC in resectable gastric cancer, should be discouraged. Restaging—comparing the CT stage at diagnosis (rTNM) to the pathologic stage following NAC (ypTNM)—is suggested instead as a useful method to evaluate response, defying the inherent challenge of baseline CT staging.

## Figures and Tables

**Figure 1 cancers-15-02318-f001:**
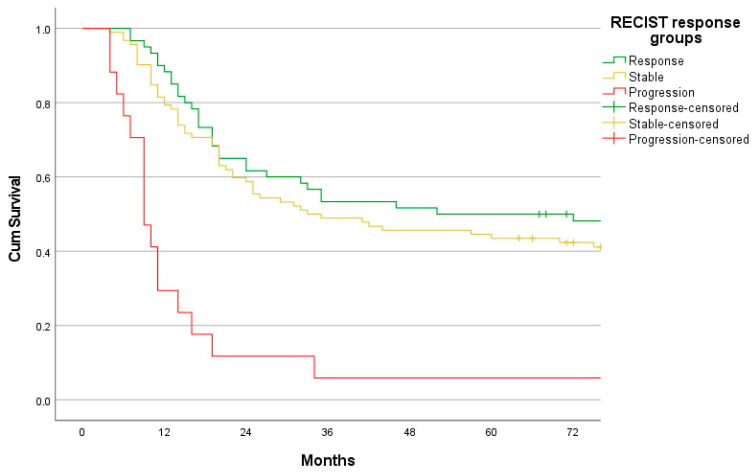
Overall survival stratified on the basis of the RECIST response mode, *n* = 169. Response (*n* = 60), stable disease (*n* = 92), and progression (*n* = 17). Response vs. stable disease log-rank *p* = 0.570.

**Figure 2 cancers-15-02318-f002:**
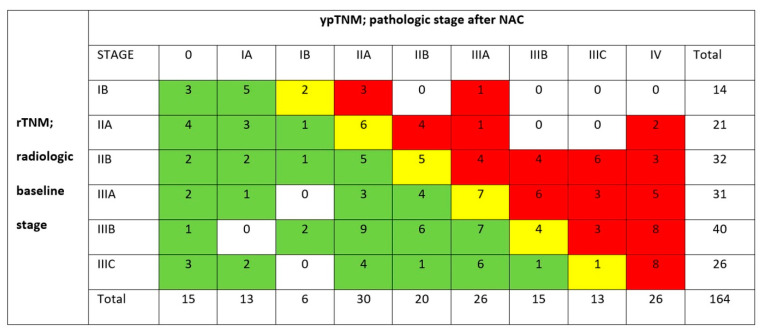
Cross-tabulation of the CT stage at diagnosis (rTNM) vs. the definitive pathological stage (ypTNM), *n* = 164. Green: response, downstaged (*n* = 78); yellow: stable disease (*n* = 25); red: progression, upstaged (*n* = 61).

**Figure 3 cancers-15-02318-f003:**
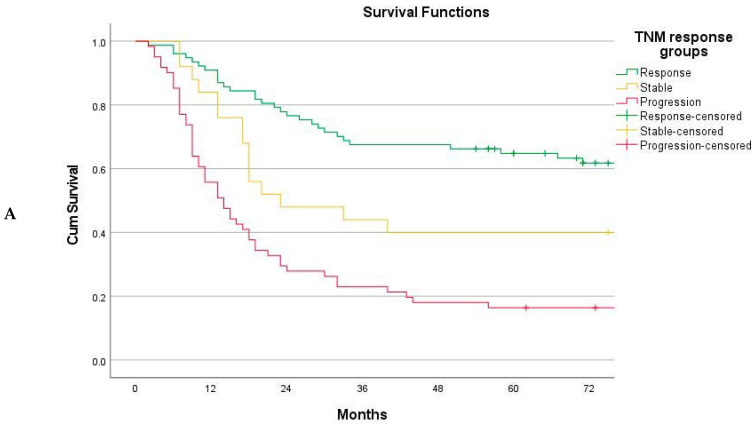
(**A**) Overall survival stratified on the basis of the TNM response mode, *n* = 164. Response (*n* = 78), stable disease (*n* = 25), and progression (*n* = 61). Global log-rank *p* < 0.001. Response vs. stable *p* = 0.012. Stable vs. progression *p* = 0.020. (**B**)*:* Overall survival based on TNM response mode. Disease progression further subdivided into progression maintaining M0 status (*n* = 35) and progression to metastatic disease M+ (*n* = 26). Stable disease vs. progression maintaining M0, log-rank *p* = 0.512.

**Figure 4 cancers-15-02318-f004:**
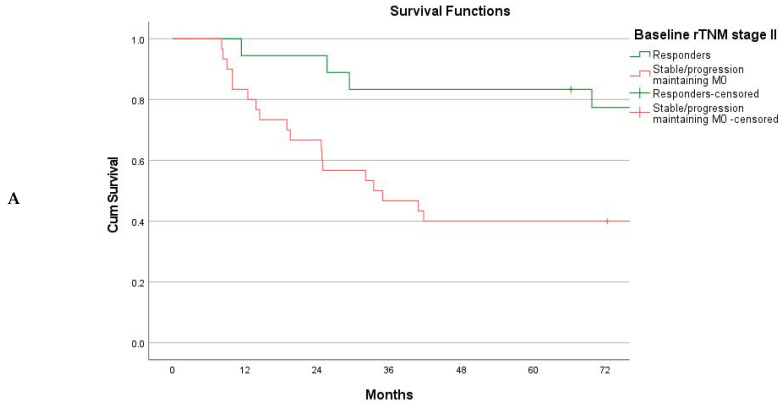
(**A**) Overall survival for baseline rTNM stage II stratified on the basis of the TNM response mode, *n* = 48. Responders (*n* = 18) vs. stable/progression maintaining M0 status (*n* = 30), log-rank *p* = 0.007. (**B**) Overall survival for baseline rTNM stage III stratified on the basis of the TNM response mode, *n* = 76. Responders (*n* = 52) vs. stable/progression maintaining M0 status (*n* = 24), log-rank *p* < 0.001. (**C**) Baseline rTNM stage III responding to NAC (*n* = 52). Overall survival stratified on the basis of down-staging by one or two tiers (*n* = 27) and by three or more tiers (*n* = 25). One/two tiers vs. three or more tiers (log-rank *p* = 0.006).

**Table 1 cancers-15-02318-t001:** Clinical variables for patients with resectable gastric cancer in 2007–2016 receiving neoadjuvant chemotherapy and available for TNM response evaluation, cross-tabulated with the response mode (*n* = 164).

	Total *n* = 164	Response *n*= 78	Stable *n* = 25	Progression *n* = 61	[M0/M+]	*p*-Value *
**Age category**						0.592
<60 years	47 (29%)	18 (38%)	9 (19%)	20 (43%)	[10/10]	
60–70 years	73 (45%)	37 (51%)	9 (12%)	27 (37%)	[17/10]	
>70 years	44 (27%)	23 (52%)	7 (16%)	14 (32%)	[8/6]	
**Gender**						0.156
Male	114 (70%)	59 (52%)	18 (16%)	37 (32%)	[19/18]	
Female	50 (30%)	19 (38%)	7 (14%)	24 (48%)	[16/8]	
**Tumor location**						0.097
Cardia	62 (38%)	33 (53%)	9 (15%)	20 (32%)	[12/8]	
Corpus	32 (20%)	17 (53%)	4 (13%)	11 (34%)	[9/2]	
Antrum	51 (31%)	25 (49%)	9 (18%)	17 (33%)	[10/7]	
Diffuse	19 (12%)	3 (16%)	3 (16%)	13 (68%)	[4/9]	
**rTNM**						0.065
Stage Ib	14 (9%)	8 (57%)	2 (14%)	4 (29%)	[4/0]	
Stage IIa/b	53 (32%)	18 (34%)	11 (20%)	24 (45%)	[19/5]	
Stage IIIa	31 (19%)	10 (32%)	7 (22%)	14 (45%)	[9/5]	
Stage IIIb	40 (24%)	25 (63%)	4 (10%)	11 (27%)	[3/8]	
Stage IIIc	26 (16%)	17 (65%)	1 (4%)	8 (31%)	[0/8]	
**Lauren classification**						<0.001
Diffuse	68 (41%)	17 (25%)	13 (19%)	38 (56%)	[22/16]	
Intestinal	70 (43%)	44 (63%)	9 (13%)	17 (24%)	[9/8]	
Mixed	26 (16%)	17 (65%)	3 (12%)	6 (23%)	[4/2]	
**MSI status**						0.392
MSI-H	8 (5%)	4 (50%)	0	4 (50%)	[2/2]	
MSS/MSI-L	148 (90%)	72 (49%)	25 (17%)	51 (34%)	[33/18]	
Unknown	8 (5%)	2 (25%)	0	6 (75%)	[0/6]	
**Number of NAC cycles delivered**						0.884
<3	23 (14%)	12 (52%)	3 (13%)	8 (35%)	[6/2]	
3	141 (86%)	66 (46%)	22 (16%)	53 (38%)	[29/24]	

* Chi-square monovariable analysis.

**Table 2 cancers-15-02318-t002:** Ordinal cumulative logistic regression with TNM response mode as the dependent variable, and entering age, gender, tumor location, Lauren histological type, and disease stage at diagnosis as the explanatory variables *.

	OR (Odds Ratio)	95% CI	*p*-Value
**Age at diagnosis**	1.01	0.98–1.04	0.528
**Gender**			
Male	1.43	0.61–3.35	0.405
**Tumor location**			
Diffuse	1		
Cardia	2.77	0.85–9.06	0.091
Corpus	5.18	1.42–18.75	0.012
Antrum	7.56	2.09–27.30	0.002
**Lauren classification**			
Diffuse	1		
Intestinal	4.43	1.98–9.89	<0.001
Mixed	5.59	2.01–15.50	<0.001
**Baseline disease stage (rTNM)**			
Stage IB	1		
Stage IIA/B	0.58	0.16–2.09	0.411
Stage IIIA	0.50	0.13–1.98	0.326
Stage IIIB	2.03	0.51–7.99	0.313
Stage IIIC	1.85	0.41–8.32	0.424

* Parallel lines assumption *p* = 0.946, Pearson’s overall goodness of fit *p* = 0.17, Nagelkerke R^2^ = 0.282.

## Data Availability

The datasets generated and analyzed during the current study are not publicly available due to hospital policy, but they are available from the corresponding author upon reasonable request. This study used data from the Cancer Registry of Norway. The interpretation and reporting of these data are the sole responsibility of the authors, and no endorsement by the Cancer Registry of Norway is intended nor should be inferred.

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
