# Peer review of "Response Evaluation after Neoadjuvant Chemotherapy for Resectable Gastric Cancer"

_cancers, 2023, doi:10.3390/cancers15082318_

Round 1

Reviewer 1 Report

The study investigates response evaluation methods following neoadjuvant chemotherapy for resectable gastric cancer. The ability to stratify patients into different survival rate subsets is important. The study compares a strict radiological method using RECIST (downsizing) with a composite radiological/pathological method comparing initial radiological TNM stage to pathological ypTNM stage (downstaging). Results show that downsizing is not a reliable method for evaluating response to neoadjuvant chemotherapy in gastric cancer, while TNM re-staging using the radiological/pathological method appears to be valuable for clinical use. The study suggests that TNM re-staging can stratify patients into different survival rate subsets based on response mode.

  The intro and methods sections were explained with sufficient detail. The discussion section provides a comprehensive analysis of the study's findings, highlighting the strengths and limitations of the methodology used. The authors identify several shortcomings in the methodology used in the study. Overall, this is well written and based on my google search there is not much written about this topic and I would consider this novel, important work.

Author Response

Response to reviewer 1

Thank you very much for the encouraging comments, and appreciation of our work as novel and important.

Reviewer 2 Report

Authors conclusions on the present study seem to be a bit exaggerated, when based on approximately 150 patients who underwent multimodal treatment with non-updated chemotherapy protocol (ECX/EOX instead of FLOT). Nonetheless, the results presented bring more insight into the current treatment of GC patients in Europe.

The manuscript could be improved with a broader discussion on the aspect of nodal staging (also in the context of NAC) and evaluating the basic surgical outcomes of the operated patients, since only R0 rate is provided.

Author Response

Response to reviewer 2:

Thank you for constructive criticism and comments.

Point 1: Authors conclusions on the present study seem to be a bit exaggerated, when based on approximately 150 patients who underwent multimodal treatment with non-updated chemotherapy protocol (ECX/EOX instead of FLOT). 

Response 1: We agree with this comment and have rephrased the sentence related to this topic in the abstract (page 1 line 42-45). We have further rephrased our conclusion to state that “Restaging, comparing CT-stage at diagnosis (rTNM) to pathologic stage following NAC (ypTNM) is suggested as a useful method to evaluate response, defying the inherent challenge of baseline CT-staging”. 

Point 2: The manuscript could be improved with a broader discussion on the aspect of nodal staging (also in the context of NAC) and evaluating the basic surgical outcomes of the operated patients, since only R0 rate is provided.

Response 2: We agree on the comment regarding nodal staging. The inherent limitations of CT to detect lymph node metastases at baseline is a challenge, with nodes deemed negative if size is < 10 mm, although with some later recognized to harbour cancer cells (1). However, lowering any diameter threshold would significantly reduce specificity.  In a systematic review EUS, CT, and MRI were equivalent in establishing T stage. Neither EUS, MRI nor PET could establish N stage with any higher precision (2, 3). This has now been broader discussed in our manuscript (page 11 line 327-332). As of radiological nodal staging following NAC, we consider this of limited relevance for this particular study (page 11 line 281-285); it is, however, extensively discussed in a previously published work from our group (4)

Regarding aspects of the surgical outcome of the operated patients, we are not sure what the reviewer is referring to?  Numbers provided on surgery are not limited to patients receiving radical resections (R0 and R1), but also include patients with R2 resections and explorative laparotomies (page 4 line 168-172, Figs. 2, 3a and 3b).  Please advise, if further detailing is wanted.

  1. Saito T, Kurokawa Y, Takiguchi S, Miyazaki Y, Takahashi T, Yamasaki M, et al. Accuracy of multidetector-row CT in diagnosing lymph node metastasis in patients with gastric cancer. Eur Radiol. 2015;25(2):368-74.
  2. Kwee RM, Kwee TC. Imaging in local staging of gastric cancer: a systematic review. J Clin Oncol. 2007;25(15):2107-16.
  3. Kwee RM, Kwee TC. Imaging in assessing lymph node status in gastric cancer. Gastric cancer : official journal of the International Gastric Cancer Association and the Japanese Gastric Cancer Association. 2009;12(1):6-22.
  4. Sandø AD, Fougner R, Grønbech JE, Bringeland EA. The value of restaging CT following neoadjuvant chemotherapy for resectable gastric cancer. A population-based study. World J Surg Oncol. 2021;19(1):212.

Reviewer 3 Report

In this article, the authors evaluated response of NAC for the patients with locally advanced gastric cancer comparing downsizing and downstaging. To evaluate downstaging, the authors utilized a novel definition of “response mode”. Finally, they concluded that downstaging, rather than down size, is more valuable methods in clinical practice.

This study has methodological problems as follow.

1) Was the primary lesion measured as target lesion in this study? In the results section there is a statement that the primary lesion was measured (page4,line164-165).

 In RECIST, the size of primary tumor is not defined as mesurable. Assessment of primary tumor size in gastric cancer on CT images is not a common method. Rationale for assessing changes in tumor size on CT images needs to be provided.

2) The criteria for nodal diagnosis in CT imaging are clearly stated, but no mention is made of the criteria for tumor depth diagnosis in CT imaging. The diagnostic criteria for tumor depth diagnosis need to be clarified.

3) I can not understand the contents of the results section, especially 3.1; Response evaluation by downstaging. What is meant by “carcinomatosis”.

In addition, I consider problematic points in this study as follow.

1) How does stratifying patients by downstaging after NAC change the treatment strategy?

Author Response

Response to reviewer 3:

Thank you for constructive criticism and comments.

As the quality of the English language has been marked as difficult to understand/incomprehensible, the manuscript was send for proofreading by the MDPI language editing serviced suggested.

Point 1: Was the primary lesion measured as target lesion in this study? In the results section there is a statement that the primary lesion was measured (page4,line164-165). In RECIST, the size of primary tumor is not defined as mesurable. Assessment of primary tumor size in gastric cancer on CT images is not a common method. Rationale for assessing changes in tumor size on CT images needs to be provided.

Response 1: This is an important remark.  RECIST was developed to assess drug activity  in phase II trials, using downsizing as a measure of response to cytotoxic treatment in solid tumours/metastases/nodes. It was never intended for clinical decision making, nor envisioned to translate response into differences in survival rates, or to be used in a neoadjuvant setting (5). Still, as evidenced in the present paper, response evaluation using the RECIST criteria or similar formats is often encountered for both gastric cancers and/or in a neoadjuvant setting (6-9). When made aware of this, we addressed the RECIST working group during the autumn of 2021 with two questions:  Question 1: Is RECIST validated for use in response evaluation in gastric cancer, as the stomach is not a parenchymatous organ?  In our research project all patients have a measurable tumour in the stomach (as well as some also with nodal target lesions, i.e. lymph nodes > 15 mm short axis). Question 2: Is it correct  to refer to the main tumour in the stomach as a target lesion?                                                                                                                                                                                              Response from the RECIST Working Group:RECIST was assessed for response evaluation of advanced or metastatic solid tumors in general in the context of clinical trials. It has however started to lead a life of its own, as illustrated by your research. For instance, RECIST has not been developed for the neo-adjuvant setting even though it is often used. So I cannot really say that RECIST was validated for your specific situation. If you have measurable primary which is not constrained in its development due to its location, i.e. you can measure a longest diameter and you would be able to demonstrate a 20% increase resulting in a PD, it can be used (5)”.

The expanded (inappropriate?) use of RECIST taking place as acknowledged by the RECIST working group, both regarding the neoadjuvant setting and treating the gastric primary as a measurable target lesion, was the very reason why we wanted to investigate the abilities and limitations of RECIST (downsizing) as method of response evaluation following NAC in a Western cohort. Some studies using RECIST in gastric cancers only measured the change in nodal target lesions to access response, excluding the gastric primary.  To be classified as a nodal target lesion, a lymph node to be ≥15 mm. Applying this to the present study left a mere 10% of the patients with a target lesion to evaluate. Hence, the gastric primary was treated as a non-nodal target lesion, in line with the developing (inappropriate?) trend, as acknowledged by the RECIST Working Group.  

Point 2:

The criteria for nodal diagnosis in CT imaging are clearly stated, but no mention is made of the criteria for tumor depth diagnosis in CT imaging. The diagnostic criteria for tumor depth diagnosis need to be clarified.

Response 2: We fully agree with this sentiment, and have included this in the methods section 2.4 (page 3 line 141-145.)

Point 3: I can not understand the contents of the results section, especially 3.1; Response evaluation by downstaging. What is meant by “carcinomatosis”

Response 3: Both section 3.1 Response evaluation by downsizing, and section 3.2 Response evaluation by downstaging has been rephrase to make it more clear. Regarding  “carsinomatosis” this has now been clarified to “peritoneal carcinomatosis”. As defined by RECIST the appearance of any new lesions (M+, including peritoneal carcinomatosis) is considered progression. Please advise, if further detailing is wanted.

Point 4: How does stratifying patients by downstaging after NAC change the treatment strategy?

Response 4: This is a pivotal question, in our opinion not restricted to this paper nor to this particular method of response evaluation. A common denominator of all methods, whether based on radiology, pathology, or both as in the present paper, is a sense of post hoc evaluation. Hence, what is the purpose? We have tried to address this both in the simple summary (page 1 line14-16) and in the discussion (page 11 line 320-323), concluding with response evaluation as both purposeful and important.

  1. RECIST Workning Group. RECIST. 2022.
  2. Achilli P, De Martini P, Ceresoli M, Mari GM, Costanzi A, Maggioni D, et al. Tumor response evaluation after neoadjuvant chemotherapy in locally advanced gastric adenocarcinoma: a prospective, multi-center cohort study. J Gastrointest Oncol. 2017;8(6):1018-25.
  3. Kurokawa Y, Shibata T, Ando N, Seki S, Mukaida H, Fukuda H. Which is the optimal response criteria for evaluating preoperative treatment in esophageal cancer: RECIST or histology? Ann Surg Oncol. 2013;20(9):3009-14.
  4. Blank S, Lordick F, Bader F, Burian M, Dobritz M, Grenacher L, et al. Post-therapeutic response evaluation by a combination of endoscopy and CT scan in esophagogastric adenocarcinoma after chemotherapy: better than its reputation. Gastric cancer : official journal of the International Gastric Cancer Association and the Japanese Gastric Cancer Association. 2015;18(2):314-25.
  5. Tang X, He Q, Qu H, Sun G, Liu J, Gao L, et al. Post-therapy pathologic tumor volume predicts survival in gastric cancer patients who underwent neoadjuvant chemotherapy and gastrectomy. BMC Cancer. 2019;19(1):797.